# Molecular Characterization of *Toxoplasma gondii* in Cats and Its Zoonotic Potential for Public Health Significance

**DOI:** 10.3390/pathogens11040437

**Published:** 2022-04-04

**Authors:** Mian Abdul Hafeez, Muntazir Mehdi, Faiza Aslam, Kamran Ashraf, Muhammad Tahir Aleem, Abdur Rauf Khalid, Adeel Sattar, Syeda Fakhra Waheed, Abdulaziz Alouffi, Omar Obaid Alharbi, Muhammad Abu Bakr Shabbir, Umer Chaudhry, Mashal M. Almutairi

**Affiliations:** 1Department of Parasitology, University of Veterinary and Animal Sciences, Lahore 54000, Pakistan; muntazir321mehdi@gmail.com (M.M.); kashraf@uvas.edu.pk (K.A.); 2Department of Pathology, University of Veterinary and Animal Sciences, Lahore 54000, Pakistan; faizadr@yahoo.com (F.A.); fakhra.waheed@uvas.edu.pk (S.F.W.); 3MOE Joint International Research Laboratory of Animal Health and Food Safety, College of Veterinary Medicine, Nanjing Agricultural University, Nanjing 210095, China; dr.tahir1990@gmail.com; 4Department of Livestock and Poultry Production, Faculty of Veterinary Sciences, Bahauddin Zakariya University, Multan 60000, Pakistan; abdurrauf@bzu.edu.pk; 5Department of Pharmacology and Toxicology, University of Veterinary and Animal Sciences, Lahore 54000, Pakistan; adeel.sattar@uvas.edu.pk; 6King Abdulaziz City for Science and Technology, Riyadh 12354, Saudi Arabia; asn1950r@gmail.com; 7Department of Pharmacology and Toxicology, College of Pharmacy, King Saud University, Riyadh 11451, Saudi Arabia; pharmdomar@hotmail.com; 8Institue of Microbiology, University of Veterinary and Animal Sciences, Lahore 54000, Pakistan; abubakr.shabbir@uvas.edu.pk; 9Department of Veterinary Epidemiology and Public Health, School of Veterinary Medicine, University of Surrey, Surrey GU27XH, UK; u.chaudhry@surrey.ac.uk

**Keywords:** toxoplasmosis, seroprevalence, PCR, phylogenetic analysis

## Abstract

Toxoplasmosis is a globally distributed disease of warm-blooded animals. It is caused by the opportunistic parasite *Toxoplasma gondii* (*T. gondii*). One-third of the global human population is believed to be infected with *T. gondii*. Cats serve as final host of *T. gondii* and are the main source of contamination of soil and water. This study aimed to detect genotypes of *T. gondii* in cats. Fecal samples (n = 400) were collected from districts of South Punjab (Khanewal and Sahiwal), and were processed by polymerase chain reaction (PCR) followed by sequencing and phylogenetic analysis. The obtained oligonucleotide sequences (*T. gondii*) were submitted to the GenBank database, and the evolutionary tree was constructed using MEGA-X software. Seven fecal samples (3.5%) from cats were positive. Five out of thirteen fecal samples (38.46%) found to be positive for *T. gondii* with microscopy were confirmed by PCR. After phylogenetic analysis with 3 clonal types and atypical strains, isolates of *T. gondii* in current study were more closely linked to a typical strain (AF249696). Besides genotyping from cats, seroprevalence from humans and ruminants is still considered to be the best and easiest way to identify the *Toxoplasma*. Blood samples were collected from sheep and goats (n = 2000 each), and human blood samples (n = 400) were collected from the same vicinity. Seroprevalence was determined using a commercial enzyme-linked immunosorbent assay (ELISA) kit. In Khanewal, the blood samples of 292 goats (29.2%) and 265 sheep (26.5%), and 6 fecal samples from cats (3%) were positive. Out of 200 human blood samples, 52 were positive, with a seroprevalence of 26%. In the Sahiwal district, the blood samples from 49 humans, 235 sheep and 348 goats were positive, with seroprevalence of 24.5%, 23.5% and 34.8%, respectively. The present study revealed the current circulating genotype of *T. gondii* from cats in the districts Khanewal and Sahiwal and the seroprevalence of the organism in small ruminants and humans living in the same vicinity. Further genotype analyses of the organism from ruminants and humans are needed.

## 1. Introduction

*Toxoplasma gondii* is a common protozoan parasite of zoonotic significance that causes serious disease in small ruminants and mammals, including humans [1,2]. As one of the five parasitic diseases targeted for public health action and prevention, this malady has become a priority (Centers for Disease Control and Prevention (CDC), 2013). Toxoplasmosis is a cause of communal infection among caprine animals around the globe [3,4]. It is known to cause reproductive miscarriage [5]. This may result in huge economic losses due to abortion or the birth of weak offspring in food animals [6,7]. The infection rate varies widely from herd to herd due to inbreeding; an average rate of 30% has been recorded. The high prevalence of toxoplasmosis among domesticated animals such as cattle, sheep and goats can be an important cause of disease transmission to humans [8].

Toxoplasmosis is a highly zoonotic disease that is instigated by *T. gondii*, a food-borne pathogen [9,10]. Human toxoplasmosis is closely linked with the use of raw or undercooked food items [11]. Meat obtained from domestic animals containing *T. gondii* is considered an important source of infection in humans.

Efforts have been made to isolate the *T. gondii* tachyzoites from the saliva, urine, vaginal mucosa and nasal secretions of infected food animals [12]. Consumption of products, including milk, from diseased food animals stands as an imperative cause of human infection, which is of concern due to a rise in the consumption of sheep and goat’s milk among children with an allergy to cow’s milk [13]. Approximately 30% of the human population is chronically infected with *T. gondii* [14].

Seroprevalence at childbearing age in women has been shown to range from 4 to 100%. During pregnancy, infection rates vary (between 1 to 310 in 10,000 pregnancies) according to the locality of the pregnant population, for example, Europe and the USA [15]. Presently, there is no gold standard test for the diagnosis of toxoplasmosis. Different methods have been recommended to analyze infected populations based on the organs affected [16]. The best approach is indirect hemagglutination [17], which can be used to detect this disease in animals with the help of readily available kits.

Despite the economic and zoonotic significance, as well as the high seroprevalence of this pathogen, little or no research has been conducted in Pakistan. This project was undertaken to persuade assessment of the seroprevalence of toxoplasmosis in sheep, goats and humans in the populated districts of South Punjab, and the molecular characterization of *T. gondii* from cats. The proposed study provides solutions to address the dilemma between animal-friendly production and safety regarding *T. gondii* as a food pathogen with a high health impact.

## 2. Materials and Methods

### 2.1. Study Area and Sample Collection

The study comprised two districts of South Punjab (Khanewal and Sahiwal) (Figure 1). Samples were collected and processed between May and November 2020. A total of 2400 blood samples were taken: 1000 goats and sheep each; 200 from humans in different localities from each of the mentioned districts. Approximately 5 mL of blood was collected from the above-mentioned species for serum separation. Samples were collected rando mLy from these animals through appropriate methods and were transported to the immuno-parasitology laboratory, Department of Parasitology, UVAS, Lahore. In addition, 200 fecal samples from cats were also collected. The presence of cats and their accessibility to animal feed and water was observed within each zone.

### 2.2. Seroprevalence Determined Using ELISA

Blood samples were centrifuged at 2300× *g* for 10 minutes and the serum was separated [18]. Immunoglobulin G (IgG) antibodies were detected and separated using commercial indirect ELISA as described by [19]. The optical densities (ODs) were calculated through use of the Microplate Reader RT-6000 (Devrim limited, Romford, UK). The samples that revealed OD values >11 were considered to represent positive samples as per the manufacturer’s instructions for results interpretation.

### 2.3. Fecal Examination

A direct smear of feces was prepared and analyzed through use of an Olympus CX21 microscope. Oocysts were identified in cat fecal samples. Flotation/sedimentation techniques were used for the identification of oocysts. Collected un-sporulated oocysts were placed in 2.5% potassium dichromate (1:5) for sporulation. After five days, sporulated oocysts were observed under a compound microscope at 40×. Oocysts were counted using the McMaster technique. Washed oocysts were subjected to excystation using glass beads [20]. The sporozoites released from oocysts suspensions were subjected to DNA extraction.

### 2.4. Molecular Identification

DNA was extracted from the fecal samples using QIAamp Fast DNA Stool Mini Kit (Qiagen, Germany) for molecular detection as per the manufacturer’s instructions. PCR was performed to amplify the COX1 gene with the use of the universal primers, 400F (5′-GGDTCAGGTRTTGGTTGGAC-3′) and 1202R (5′-CCAAKRAYHGCACCAAGAGATA-3′). The PCRs were optimized by varying the concentrations of the DNA template [21]. Specific forward primers (5′-CCTGGTGTCTCTTCAAGCGT-3′) and reverse primers (5′-AAAGGAGAATGAGCGCACGA-3′) of the SAG2 gene with amplicon size 529 bp were used [22]. The PCRs were performed using the following conditions: initial denaturation at 95 °C for two minutes; followed by 35 cycles comprising of denaturation at 95 °C for 30 seconds; annealing at 59 °C for 30 seconds; extension at 72 °C for two minutes; and final extension at 72 °C for 10 minutes. Agarose gel (1.5%) was prepared and stained with ethidium bromide in order to analyze the PCR-generated amplicons. Gel electrophoresis was performed at 113 V, 230 mA for 35 minutes in order to visualize the bands in the gel documentation system (Bio-Rad Laboratories, Hercules, CA, USA).

### 2.5. Sequencing and Phylogenetic Analysis

*T. gondii* amplicons were arbitrarily selected for molecular analysis, purified with Gene JET Gel Extraction Kit (Thermo Fisher Scientific, Waltham, MA, USA) according to the manufacturer’s instructions and subjected to Sanger’s sequencing method on an Applied BioSystems 3130 automated DNA sequencer. Phylogenetic analysis was performed using MEGA X software (Version: 11) through the neighbor-joining method with 1000 bootstraps to accomplish several sequence alignments. A phylogenetic tree of *T. gondii* SAG2 was constructed and compared with sequences that were already available in GenBank.

### 2.6. Statistical Analysis

Seroprevalence was assessed using the chi-square test. The *p*-values for results from both districts were found to be significant (*p* ≤ 0.05). Detection of *T. gondii* through the use of both the microscope and PCR was also found to be statistically significant.

## 3. Results

### 3.1. Seroprevalence of Toxoplasmosis in Small Ruminats and Humans

In Khanewal, 292 goats (29.2%), 265 sheep (26.5%) and 52 humans (26%) tested positive for *T. gondii* antibodies. In the Sahiwal district, 384 goats (38.4%), 235 sheep (23.5%) and 49 humans (24.5%) tested positive. The antibodies that were detected for seropositive samples by ELISA in small ruminants and human samples are presented in Table 1. The seroprevalence of toxoplasmosis in sheep, goats and humans in Sahiwal and Khanewal districts were found to be statistically significant (*p* ≤ 0.05) using the Chi-square test in SPSS.

### 3.2. Examination of Fecal Samples

Through the use of microscopy, six (3%) and seven (3.5%) cat fecal samples were found to be positive in Khanewal and Sahiwal, respectively (Table 2). Unsporulated oocysts were spherical or somewhat round in shape. Their size ranged from 10 µm to 14 µm, measured by micrometry.

### 3.3. Molecular Detection by PCR

Microscopically positive fecal samples were confirmed by PCR. Five out of thirteen samples from both districts (38.46%) of *T. gondii* oocysts under the microscope were confirmed through PCR. Partial amplification of repetitive 800 bp and 529 bp sections are shown in Figure 2 and Figure 3, with universal and specific primers, respectively. The prevalence of *T. gondii* oocysts according to microscopy and PCR in the cats was analyzed by chi-square tests and was found to be statistically significant (*p* ≤ 0.05).

### 3.4. Sequencing and Phylogenetic Analysis

The SAG2 gene of three current isolates was sequenced and compared with published sequences from all clonal types of *T. gondii* (Table 3). Clustal W analysis and bootstrapping were performed to enable the genotypic comparison of three isolates from cat fecal samples with published strains of *Toxoplasma* in GenBank. The current study isolates were found to be closer to the atypical strain AF249696 (Figure 4). Representative strains of each genotype from type I, II or III were taken after Clustal W analysis. Thus, it was found that our three isolates were near to atypical.

## 4. Discussion

In this study, ELISA was performed to estimate the seroprevalence of *T. gondii* in ruminants and humans. The findings of the present study provided evidence of toxoplasmosis in ruminants, humans and cats. The seroprevalence of *T. gondii* in humans from both districts was in agreement with the findings of [23], in upper Myanmar [24] and Iran [25]. Higher seroprevalence of *T. gondii* has been reported in Egypt [26], Iran [27], Northern Iraq [28], Egypt [29], Sudan [30], North East India [31] and Saudi Arabia [32] than in current study. However, lower seroprevalence in humans has been reported in Khyber Pakhtunkhwa, Pakistan [33], South-Eastern China [34], India [35], West Africa [36], Pothwar region, Northern Punjab, Pakistan [37], Ghana [38] and Algeria [39].

The present study revealed a higher seroprevalence of toxoplasmosis in both South Punjab districts (Khanewal and Sahiwal). These findings may be due to the warm and humid environment in Southern Punjab, which is favorable for the survival of *T. gondii* oocysts. Different rates of seroprevalence were recorded at different locations, which may be due to changes in the environment, sanitary conditions and management practices.

The prevalence of *T. gondii* oocysts in the present study was similar to the prevalence of *T. gondii* reported in Lahore, Pakistan (2.3%) [40]. Similar prevalence has also been reported in China (2.78%) [41] and Iran (2.56%) [42]. The prevalence was higher in Indonesia (6.81%) [22] and Kenya (7.8%) than in the present study [1]. Microscopy is not a reliable way to screen for oocyst shedding in cats because this method of oocyst identification lacks sensitivity and specificity. Infective oocysts can only be detected in cats infected with *T. gondii*. Antibodies and tissue cysts remain in the body for a long period of time, whereas oocysts persist for three weeks after infection [43].

In the current study, the prevalence of toxoplasmosis in cats was in line with the prevalence reported in Yogyakarta, Indonesia [22]. The prevalence reported in this study was higher than that reported in Izmir, Turkey (15.34%) [44]. Lower prevalence of *T. gondii* in domestic cats has been reported in Brazil (3.74%) [45] and Poland (2.4%) [46]. A higher prevalence of 47.2% has been reported in South Korea [47] and Southern Thailand (61.5%) [48]. The low prevalence of toxoplasmosis reported in domestic cats may be because these cats are mostly kept indoors [49]. Pet cats are usually prevented from eating raw or poorly cooked meat and have little chance of contact with wild animals.

Sequencing and phylogenetic results revealed that atypical SAG2 was the predominant genotype of *T. gondii* circulating in the study area. Limited research has been conducted in Pakistan to identify and characterize the molecules of *T. gondii*. Previous studies have indicated that the *T. gondii* SAG2 gene consists of three chief clonal lineages, nominated as types I, II and III, which are important in Europe and the USA [50], whereas the atypical genotype is dominant in South America and Asia [51,52]. In the current research, the phylogenetic approach revealed the SAG2 atypical strain through nucleotide sequence analysis. Although the lack of other genotypes in this study was consistent with previous findings, some studies have also established types II or III as the main genotypes [53]. In previous studies, the SAG2 atypical strain was also identified from Pakistan [54].

Atypical *T. gondii* genotypes were found to be associated with a number of severe cases of toxoplasmosis in immunocompetent individuals [55]. Atypical genotypes can develop when a cat ingests prey infected with *T. gondii* of more than one clonal type, followed by sexual recombination in the gut of the cat which can result in progeny representing a mixture of the two parental genotypes [56]. It is most likely that these atypical genotypes are the result of sexual recombination in cats [57]. Type 1 has been reported in China [58] and Iran [59]. These conflicting results are probably due to the presence of various genotypes of *T. gondii* in different geographical regions. It has been shown that *Toxoplasma* strains in animals and humans have different distributions in the same geographical regions.

## 5. Conclusions

The present study revealed the current circulating genotype of *T. gondii* from cats in the districts Khanewal and Sahiwal, and the seroprevalence of the organism in small ruminants and humans living in the same vicinity. Further studies are needed to assess the genotyping of *T. gondii* in intermediate hosts (humans and small ruminants) in Pakistan.

## Figures and Tables

**Figure 1 pathogens-11-00437-f001:**
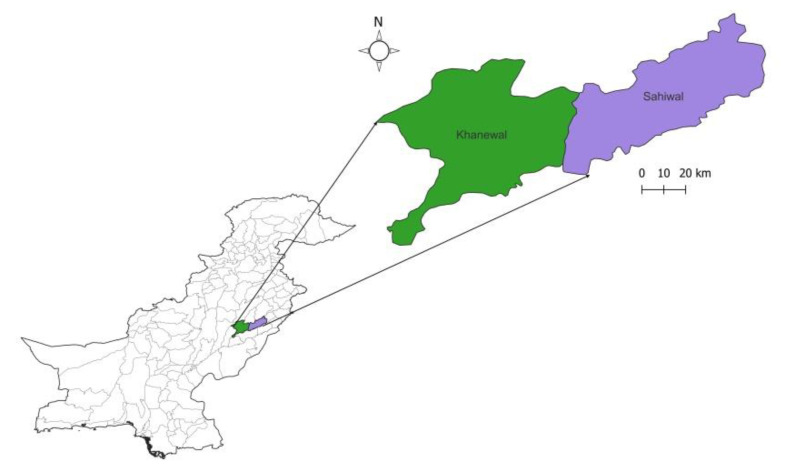
Map representing the sites where samples were collected.

**Figure 2 pathogens-11-00437-f002:**
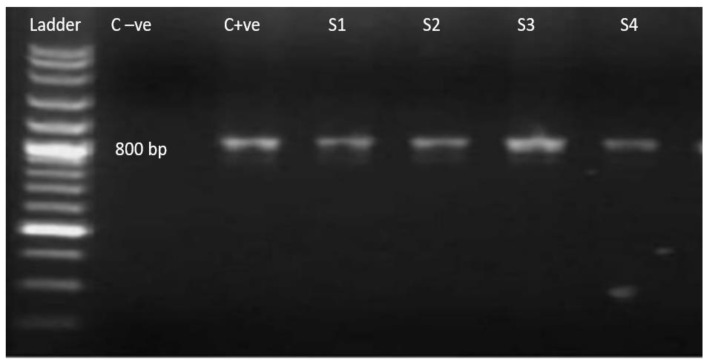
Ethidium bromide-stained agarose gel with PCR amplification 800 bp (using universal primers). Ladder: 100 bp molecular weight marker; C−ve: negative control; C+ve: positive control; S1–S4 current study isolates ensuring amplified product size.

**Figure 3 pathogens-11-00437-f003:**
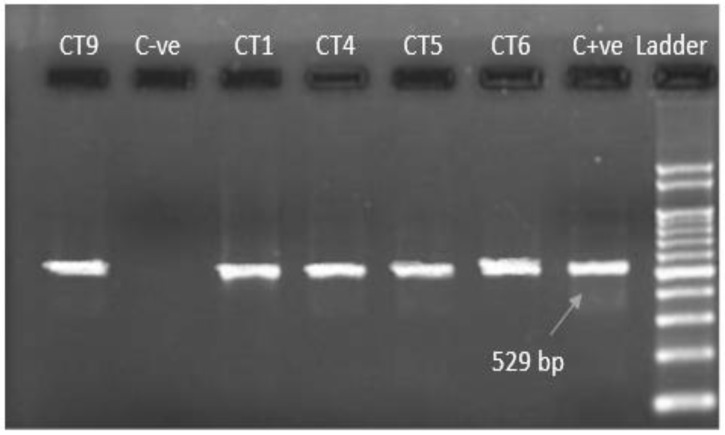
Ethidium bromide-stained agarose gel with PCR amplification 529 bp (using specific primer). C−ve: negative control; CT9, CT1, CT4, CT5, CT6: current study isolates; C+ve: positive control; Ladder: 100 bp DNA marker.

**Figure 4 pathogens-11-00437-f004:**
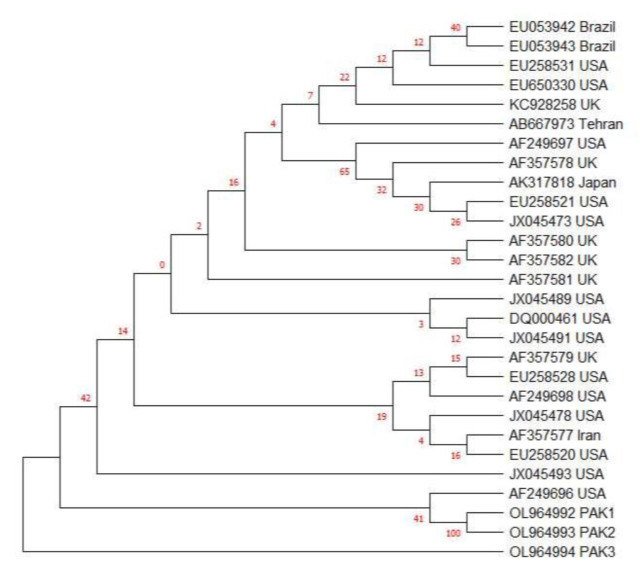
SAG2 gene-based phylogenetic analysis of 3 current isolates from cat fecal samples using the neighbor-joining method with 1000 bootstrap replicates, with the help of Tamura Nei model (29) through MEGA-X software. PAK1, PAK2 and PAK3 are current study isolates from cat feces. AF249696 is the SAG2 sequence of the atypical *T. gondii* strain.

**Table 1 pathogens-11-00437-t001:** Seroprevalence of toxoplasmosis in small ruminants and humans.

Area	Species	Total Samples	Seropositive	Seronegative	Prevalence (%)
Khanewal	Human	200	52	148	26.00
	Goat	1000	292	708	29.20
	Sheep	1000	265	735	26.50
Sahiwal	Human	200	49	151	24.50
	Goat	1000	348	652	34.80
	Sheep	1000	235	765	23.50

**Table 2 pathogens-11-00437-t002:** Prevalence of *T. gondii* in cat fecal samples (microscopy and PCR).

Area	Total Samples	Microscopy			PCR		
		Positive	Negative	Prevalence (%)	Positive	Negative	Prevalence (%)
Khanewal	200	6	194	3.00	2	4	33.33
Sahiwal	200	7	193	3.5	3	4	42.85

Pearson chi-square = 4.000, DF = 1, *p* = 0.000 (*p*-value < 0.05 is statistically significant).

**Table 3 pathogens-11-00437-t003:** SAG2 sequences along with their accession numbers retrieved from NCBI with current isolates for phylogenetic analysis.

Genotype	Accession Number	Country
**Type I**	AK317818	Japan
	EU053942	Brazil
	JX045478	USA
	EU258520	USA
**Type II**	EU258521	USA
	KC298258	UK
	EU053943	Brazil
	AF249697	USA
	AF357578	UK
	JX045473	USA
**Type III**	AF357577	Iran
	AF249698	USA
	AB667973	Tehran
	DQ000461	USA
**Atypical**	AF249696	USA
	JX045493	USA
	JX045491	USA
	JX045489	USA
	AF357582	UK
	AF357580	UK
	AF357581	UK
	EU258531	USA
	EU650330	USA
	OL964992	Pakistan
	OL964993	Pakistan
	OL964994	Pakistan

## Data Availability

The datasets generated during and/or analyzed during the current study can be find in the main text.

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
