# Peer review of "Molecular Characterization of Toxoplasma gondii in Cats and Its Zoonotic Potential for Public Health Significance"

_pathogens, 2022, doi:10.3390/pathogens11040437_

Round 1

Reviewer 1 Report

Overview of the Manuscript:  In the study, the authors aim to explore the zoonotic potential of Toxoplasma gondii in animals raised for food (i.e. goats and sheep) and in humans. The experimental approach was to collect blood of goats, sheep and human and fecal samples from cats, and test for the presence of T. gondii infection in blood via assaying seroprevalence using an ELISA and morphological assay and PCR to test for oocysts in cat feces. The PCR results from the oocysts were sequenced and used for phylogenetic analysis. Two regions in the Punjab region of Pakistan were tested. Seroprevalence in goats, sheep and humans in both regions were found to be between 23-35% and determined to statistically significant via Chi-square test. The incidence of Toxoplasma oocysts was 3-3.5% in both regions and also shown to be statistically significant via Chi-square test. Phylogenetic analysis found the strain of Toxoplasma from oocysts in cats was determined to be an atypical strain. The authors conclude these results reveal a potential impact for Toxoplasmosis and reflect an endemic threat to small ruminants and humans, with a key role of cats in the transmission of the disease.

The manuscript consists of 3 Tables and 4 Figures.

General review of the Manuscript: In this manuscript address an important issue of zoonotic potential of Toxoplasmosis in animals raised for food and to humans in Pakistan. This is an important question. However there several major problems with the manuscript including: 1) over-interpretation of the data and meaning of the results, 2) incomplete description of the how the data was obtained, 3) errors in Tables presenting the data and questionable or misinterpretation of data relating to the oocyst PCR data and 4) Mis-labeling of Figures and incomplete figure legends. These problems need to be addressed and corrected before this manuscript is acceptable for publication. There are also a few minor issues with the writing and/or grammar as noted below.

Areas of Minor and some Major Concerns (indicated by * & in bold):

Introduction

  1. lines 60-63: is infection from ‘inhalation of oocysts’ that common and if so when? This is not usually cited as a mode of infection or at least not to the same degree as the other modes of infection as cited in this sentence (i.e. eating raw or uncooked meat, etc.)
  2. lines 73-74: The phrasing ‘Till today’ is awkward and would be better phrased as ‘Presently…’
  3. lines 75-76: The statement in reference to analysis of infected populations, the authors write: ‘The best-suited approach is indirect haem-agglutination’. This is not the typical approach used in humans. I think the authors may be referring to the standard in animals. This should be clarified.

*4. Lines 81-84: The authors state ‘The aim of this study was to standardize the enzyme-linked immunosorbent assay (ELISA) test and the confirmation method by polymerase chain reaction (PCR) for the characterization of the causative agent …’. I do not see data presented in this manuscript coming anywhere near standardizing ELISA and PCR assays. Thus is a gross over interpretation of results within this manuscript and this statement needs to be re-written to more accurately reflect the data presented within and what this data actually tells us.

Materials and Methods

*1. Lines 87-98, Section 2.1 Study area and sample collection: The number of animals and human samples are indicated but there is other information on how animals and/or humans were selected for this study or demographic information such as age, sex ratios, etc. for humans and similarly, but additionally for animals, where there came from (family farms, open or free range, etc.). These conditions will affect interpretation of seroprevalence data and interpretation of how these animals were exposed to Toxoplasma and thus of importance to the data. Likewise, there is no information of where or how cats were selected for this study. Whether they were household cats that went outside, were feral or free-ranging cats, etc.

Results

*1. lines 177-189, Section Sero-biochemical analysis: It is not clear how these results are significant as the trends are distinct in goats, sheep and humans. Elaboration on this necessary or the data can simply be omitted from this manuscript as it does not appear to add anything to the analysis of the data.

*2. Lines 195-199, Section on Examination of faecal samples: There are several problems with this section. 1). a microscopic image is presented as an example of an oocysts. The image however is out of focus; the size is correct and likely these are oocysts, but an out-of-focus image not particularly convincing or reassuring.  2).  The reference to the figure is mis-labelled as Figure 2 when it should be Figure 1. 3). The math as presented in Table 3 is not correct. The total samples for Khanewal is listed as 200 as is for the Sahiwal area. The number of positive cases is 6, the number of negative cases is 194 which is 3% as reported but for Sahiwal they report 7 positive cases, 93 negative cases which would be 7% & not 3.5 as listed in Table 3. Likely the authors mean the ‘93’ to be ‘193’. However then the Table would indicate 400 cats were sampled and the authors state in the Materials and Methods (line 93) that 200 cats were sampled. Thus, the numbers in the Table are wrong and it raises the question of whether the actual percentages in cats are 6 & 7% respectively. This data needs to be corrected. 

*3. Lines 207-213, Section 3.4 Molecular detection by PCR: There are several problems with this section. 1). The authors report 5 out of 13 oocyst samples (38.46%) were positive by PCR. They refer to Table 3 but the data in Table 3 doesn’t conform to the reported 38.46% as in Table 3 which is reported separately for each region. This is confusing and the description of the data in the paper should align with the way the data is presented in the paper.  2). More significantly however, it is not clear why only 30-40% of samples positive for oocysts would be positive by PCR. I would expect closer to 90-100% of the sample to be positive. This could imply misdiagnosis of oocysts microscopically, which based on the microscopy presented in Fig. 1 is possible.  Identification of oocysts is straightforward but there could be other Coccideans that also would produce an oocyst but not Toxoplasma; this could explain the data. However, the authors offer no explanation so it is hard to discern the meaning of this PCR data relative to the oocyst data. 3). PCR results from the oocysts are shown in Figures 2 & 3; these are mis-labeled as Figures 3 & 4.

*4. Lines 213-221, Figures 2 & 3 – The figure legends are inadequate to interpret the figures. S1-S4 are not identified in Figure 2. CT1-CT6 are not identified in Figure 3. Four samples are shown in each figure but the text states 5 oocysts were confirmed by PCR (line 209) which is inconsistent with the data presented in the Figures.

Discussion

  1. lines 245-6. The authors said ‘prevalence figures were comparable with those previously reported’. What then is new or is added by this manuscript? I believe this manuscript does add relevant information, so the authors could add a 1 sentence summary of the new data that is added in this manuscript; this would help the paper.

  1. lines 247-: In this sentence the authors state these results are similar to a previous study with a reference cited in the middle of the sentence; this reference should go at the end of the sentence.

  1. lines 284-288: There is an incomplete sentence in this paragraph confusing the meaning of the paragraph. The authors also discuss some of the sero-biochemical data. I found it hard to follow the logic of the significance of the data. Improvement of the writing of this paragraph could help to clarify what the authors think is the meaning of this data. Or the data could eliminated and also this paragraph.

*4. line 306-7: The statement that the ‘PCR results reported in this study were found to be highly sensitive depending on the concentration of T. gondii oocysts in the sample’ is not supported by the data presented. No mention in the Results section were made of the PCR results relative to how many oocysts were in the sample. This also goes back to issues with Fig. 3 and the inaccuracies of the reporting of this data. This issue with interpretation of the PCR data needs to be corrected in the presentation of the data and then the meaning discussed here. From what is presented in the manuscript I do not see that the PCR data on oocysts is highly sensitive as PCR verified only 30-40% of sample identified as oocyst-positive.

Author Response

STATEMENT Thanks for your comment and suggestions. Various suggestions of the reviewer have been tackled and we have responded to each and every point of the referee is given below. Title Molecular Characterization of Toxoplasma gondii in Cats and Its Zoonotic Potential for Public Health Significance REVIEWER’S GUIDE SECTION I: Comments per Section of Manuscript Areas of Minor and some Major Concerns Introduction Infection from ‘inhalation of oocysts’ that common and if so when? This is not usually cited as a mode of infection or at least not to the same degree as the other modes of infection as cited in this sentence (i.e. eating raw or uncooked meat, etc.) As per the reviewer’s advice the said sentence has been retracted. The phrasing ‘Till today’ is awkward and would be better phrased as ‘Presently…’ The statement in reference to analysis of infected populations, the authors write: ‘The best-suited approach is indirect haem-agglutination’. This is not the typical approach used in humans. I think the authors may be referring to the standard in animals. This should be clarified. The authors state ‘The aim of this study was to standardize the enzyme-linked immunosorbent assay (ELISA) test and the confirmation method by polymerase chain reaction (PCR) for the characterization of the causative agent. I do not see data presented in this manuscript coming anywhere near standardizing ELISA and PCR assays. Thus is a gross over interpretation of results within this manuscript and this statement needs to be re-written to more accurately reflect the data presented within and what this data actually tells us. Updated Corrected “The aim of the study was to detect the seroprevalence by ELISA in goats, sheep and human beings and molecular characterization of T. gondii in cats.” Materials and Methods The number of animals and human samples are indicated but there is other information on how animals and/or humans were selected for this study or demographic information such as age, sex ratios, etc. for humans and similarly, but additionally for animals, where there came from (family farms, open or free range, etc.). These conditions will affect interpretation of seroprevalence data and interpretation of how these animals were exposed to Toxoplasma and thus of importance to the data. Likewise, there is no information of where or how cats were selected for this study. Whether they were household cats that went outside, were feral or free-ranging cats, etc. As far as the data is concerned the sampling was performed both pet and stray cats of the same vicinity with the females and sheep and goats. As far as the sampling in human beings is concerned samples were collected from the females being more prone to the Toxoplasma (causing abortions/still birth) Results Section Sero-biochemical analysis: It is not clear how these results are significant as the trends are distinct in goats, sheep and humans. Elaboration on this necessary or the data can simply be omitted from this manuscript as it does not appear to add anything to the analysis of the data As per advice of the reviewer the data is omitted A microscopic image is presented as an example of an oocysts. The image however is out of focus; the size is correct and likely these are oocysts, but an out-of-focus image not particularly convincing or reassuring. The reference to the figure is mis-labelled as Figure 2 when it should be Figure 1. 3). The math as presented in Table 3 is not correct. The total samples for Khanewal is listed as 200 as is for the Sahiwal area. The number of positive cases is 6, the number of negative cases is 194 which is 3% as reported but for Sahiwal they report 7 positive cases, 93 negative cases which would be 7% & not 3.5 as listed in Table 3. Likely the authors mean the ‘93’ to be ‘193’. However then the Table would indicate 400 cats were sampled and the authors state in the Materials and Methods (line 93) that 200 cats were sampled. Thus, the numbers in the Table are wrong and it raises the at fecal sample (n=200) were collected from each district (corrected at line 93). Figure is removed There was a calculation mistake which is now corrected. A total number of 400 feacal samples were collected. Out of 200 fecal sample collected from Khanewal only 6 (3%) were confirmed, while 7 (3.5%) were confirmed out of 200 in district Sahiwal (Corrected now table 2). There are several problems with this section. 1). The authors report 5 out of 13 oocyst samples (38.46%) were positive by PCR. They refer to Table 3 but the data in Table 3 doesn’t conform to the reported 38.46% as in Table 3 which is reported separately for each region. This is confusing and the description of the data in the paper should align with the way the data is presented in the paper. 2). More significantly however, it is not clear why only 30-40% of samples positive for oocysts would be positive by PCR. I would expect closer to 90-100% of the sample to be positive. This could imply misdiagnosis of oocysts microscopically, which based on the microscopy presented in Fig. 1 is possible. Identification of oocysts is straightforward but there could be other Coccideans that also would produce an oocyst but not Toxoplasma; this could explain the data. However, the authors offer no explanation so it is hard to discern the meaning of this PCR data relative to the oocyst data. 3). PCR results from the oocysts are shown in Figures 2 & 3; these are mis-labeled as Figures 3 & 4. PCR is most sensitive and specific diagnostic tool. Microscopic examination was used as a screening test. Higher prevalence of oocysts on microscopic examination may be due to other Coccidian species with the similar oocysts morphology. Following are the certain factors which may hinders the amplification of DNA due to which percentage was less as compared to the microscopy There could be the denaturation of extracted DNA and concentration of the DNA. The data in the table is corrected (regarding PCR and microscopy) Figures are labeled properly Fig. 2 and fig. 3 (Ethidium bromide-stained agarose gel with PCR amplification) The figure legends are inadequate to interpret the figures. S1-S4 are not identified in Figure 2. CT1-CT6 are not identified in Figure 3. Four samples are shown in each figure but the text states 5 oocysts were confirmed by PCR (line 209) which is inconsistent with the data presented in the Figures. Out of 13 positive samples on microscopy 5 were found positive by PCR. Confirmation of the positive samples was executed at different times. Discussion The authors said ‘prevalence figures were comparable with those previously reported’. What then is new or is added by this manuscript? I believe this manuscript does add relevant information, so the authors could add a 1 sentence summary of the new data that is added in this manuscript; this would help the paper. In the present study authors correlate the interaction of the small ruminants, cats and females living in the same vicinity which reflects the disease communicated from one to the other. In order to restrict the disease by creating awareness among the females. In this sentence the authors state these results are similar to a previous study with a reference cited in the middle of the sentence; this reference should go at the end of the sentence. Corrected There is an incomplete sentence in this paragraph confusing the meaning of the paragraph. The authors also discuss some of the sero-biochemical data. I found it hard to follow the logic of the significance of the data. Improvement of the writing of this paragraph could help to clarify what the authors think is the meaning of this data. Or the data could eliminated and also this paragraph. Eliminated The statement that the ‘PCR results reported in this study were found to be highly sensitive depending on the concentration of T. gondii oocysts in the sample’ is not supported by the data presented. No mention in the Results section were made of the PCR results relative to how many oocysts were in the sample. This also goes back to issues with Fig. 3 and the inaccuracies of the reporting of this data. This issue with interpretation of the PCR data needs to be corrected in the presentation of the data and then the meaning discussed here. From what is presented in the manuscript I do not see that the PCR data on oocysts is highly sensitive as PCR verified only 30-40% of sample identified as oocyst-positive. PCR is most sensitive and specific diagnostic tool. Microscopic examination was used as a screening test. Higher prevalence of oocysts on microscopic examination may be due to other Coccidian species with the similar oocysts morphology. Following are the certain factors which may hinders the amplification of DNA due to which percentage was less as compared to the microscopy There could be the denaturation of extracted DNA and concentration of the DNA. The data in the table is corrected (regarding PCR and microscopy).

Reviewer 2 Report

The overall rating of the work is high.  First of all, I agree with the authors regarding the appropriateness of conducting the study. The structure of the study itself is correct, the goals and objectives of the work were clearly presented. I have no objections to the methodology and ethical aspects. In the introduction, the authors briefly presented the current state of knowledge on the analyzed subject. Description of research methods and presented results is correct. Conclusions were drawn based on the results obtained. The discussion section is based on modern publications, confronting the obtained results with other studies. Overall, the paper is good and I think that after linguistic corrections it should be published. 

Author Response

Reviewer 2

This study, “Molecular Characterization of Toxoplasma gondii in Cats and its Zoonotic Potential for Public Health Significance Molecular Characterization of Toxoplasma gondii in Cats and its Zoonotic Potential for Public Health Significance,” performed a Toxoplasma serological survey in different hosts and isolated an atypical Toxoplasma strain circulating in felines.

While it is essential to perform studies such as this one, some improvements are necessary to solidify the presented findings:

  1. It is unclear whether the strain circulating in humans and other hosts is the same one isolated in this study. The authors need to justify this in the paper to make it cohesive.

Response: Thanks for your comment: we corrected accordingly

  1. As there is a geographical component in this study, there would be highly recommended to include a map showing where the specimens were obtained and the % of positive serum and oocysts for cats.

Response: Thanks for your comment: “corrected”

  1. Figure 1 is of low quality, and they have some writing in them. A better image is required, or this could be removed, as it doesn’t add anything to the manuscript.

Response: Thanks for your comment: “corrected”

  1. Going back to comment b), adding a map of where the oocysts were found would help, as the isolated strains seem to be different.

Response: Thanks for your comment: “added”

  1. The phylogenetic analysis is not convincing; it is based on one sequence of one PCR. There are classical differences among types 1, 2, and 3, and the authors could have added these genes to the analysis. You cannot label a strain as atypical without showing that it has chromosomes that match two or more types. The authors must show that these isolated strains have at least two significant changes that match distinct genotypes. Also, it would be great if at least a plaque assay, or ideally a replication curve, and a % bradyzoite differentiation assay. These low-cost experiments would solidify the finding and make the paper a lot better.

Response: Thanks for your comment: “corrected”

  1. Table 2 is tough to understand, and the authors have to show the data better. The numbers should be plotted (with the individual samples shown). It is unclear if this is an average of one particular sample or the whole group. If it is only one, this isn’t very sensible, and I would recommend either adding as a supplement or completely removing it.

Response: Thanks for your comment: “corrected”

Reviewer 3 Report

This study, “Molecular Characterization of Toxoplasma gondii in Cats and its Zoonotic Potential for Public Health Significance Molecular Characterization of Toxoplasma gondii in Cats and its Zoonotic Potential for Public Health Significance,” performed a Toxoplasma serological survey in different hosts and isolated an atypical Toxoplasma strain circulating in felines.

While it is essential to perform studies such as this one, some improvements are necessary to solidify the presented findings:

  1. It is unclear whether the strain circulating in humans and other hosts is the same one isolated in this study. The authors need to justify this in the paper to make it cohesive.
  2. As there is a geographical component in this study, there would be highly recommended to include a map showing where the specimens were obtained and the % of positive serum and oocysts for cats.
  3. Figure 1 is of low quality, and they have some writing in them. A better image is required, or this could be removed, as it doesn’t add anything to the manuscript.
  4. Going back to comment b), adding a map of where the oocysts were found would help, as the isolated strains seem to be different.
  5. The phylogenetic analysis is not convincing; it is based on one sequence of one PCR. There are classical differences among types 1, 2, and 3, and the authors could have added these genes to the analysis. You cannot label a strain as atypical without showing that it has chromosomes that match two or more types. The authors must show that these isolated strains have at least two significant changes that match distinct genotypes. Also, it would be great if at least a plaque assay, or ideally a replication curve, and a % bradyzoite differentiation assay. These low-cost experiments would solidify the finding and make the paper a lot better.
  6. Table 2 is tough to understand, and the authors have to show the data better. The numbers should be plotted (with the individual samples shown). It is unclear if this is an average of one particular sample or the whole group. If it is only one, this isn’t very sensible, and I would recommend either adding as a supplement or completely removing it.

Author Response

Reviewer 3

Thanks for your comment and suggestions.

Title

Molecular Characterization of Toxoplasma gondii in Cats and Its Zoonotic Potential for Public Health Significance

REVIEWER’S GUIDE

SECTION I: Comments per Section of Manuscript

1.

It is unclear whether the strain circulating in humans and other hosts is the same one isolated in this study. The authors need to justify this in the paper to make it cohesive.

The present study revealed the current circulating genotype of T. gondii from cats in the districts Khanewal and Sahiwal and the seroprevalence of the organism in the small ruminants and human living in the same vicinity. Further genotype analyses of the organism from ruminants and human need to be performed.

Seroprevalence from humans and ruminants still considered to be the best and easiest way to identify the toxoplasma.

2.

As there is a geographical component in this study, there would be highly recommended to include a map showing where the specimens were obtained and the % of positive serum and oocysts for cats.

Added

3.

Figure 1 is of low quality, and they have some writing in them. A better image is required, or this could be removed, as it doesn’t add anything to the manuscript.

Removed

 4.

5.

Going back to comment b), adding a map of where the oocysts were found would help, as the isolated strains seem to be different.

The phylogenetic analysis is not convincing; it is based on one sequence of one PCR. There are classical differences among types 1, 2, and 3, and the authors could have added these genes to the analysis. You cannot label a strain as atypical without showing that it has chromosomes that match two or more types. The authors must show that these isolated strains have at least two significant changes that match distinct genotypes. Also, it would be great if at least a plaque assay, or ideally a replication curve, and a % bradyzoite differentiation assay. These low-cost experiments would solidify the finding and make the paper a lot better.

Added

Three isolates were sequenced in present study and we have added gene sequences from three clonal types of T. gondii and atypical strains. After phylogenetic analysis with 3 clonal types and atypical strains, current study isolates of T. gondii were found more closely linked to a typical strain (AF249696).

6.

Table 2 is tough to understand, and the authors have to show the data better. The numbers should be plotted (with the individual samples shown). It is unclear if this is an average of one particular sample or the whole group. If it is only one, this isn’t very sensible, and I would recommend either adding as a supplement or completely removing it.

Removed

Round 2

Reviewer 1 Report

The authors adequately addressed my concerns.

Reviewer 3 Report

The authors have worked on the suggested changes, and I recommend that the article be accepted.